# Level of Physical Activity and Its Associated Factors among Primary Healthcare Workers in Perak, Malaysia

**DOI:** 10.3390/ijerph17165947

**Published:** 2020-08-16

**Authors:** Hazizi Abu Saad, Pei Kit Low, Rosita Jamaluddin, Huei Phing Chee

**Affiliations:** 1Department of Nutrition and Dietetics, Faculty of Medicine and Health Sciences, University Putra Malaysia, Serdang 43400, Malaysia; lowpeikit@gmail.com (P.K.L.); rositaj@upm.edu.my (R.J.); 2Sports Academy, University Putra Malaysia, Serdang 43400, Malaysia; 3Department of Allied Health Sciences, Faculty of Science, University Tunku Abdul Rahman, Kampar 31900, Malaysia; cheehp@unitar.edu.my

**Keywords:** physical activity, primary healthcare workers, sedentary behaviours, GPAQ, occupational, transport-related, leisure time, Malaysia

## Abstract

Physical inactivity and sedentary lifestyle have been linked to the occurrence of non-communicable diseases. This study’s purpose was to determine physical activity levels and sedentary behaviours among primary healthcare workers in Perak, Malaysia, as well as associated factors. A cross-sectional study was conducted at 12 health clinics in Perak, Malaysia, to determine physical activity levels, sedentary behaviours and factors associated with physical inactivity among primary healthcare workers. Each respondent completed a self-administered questionnaire relating to socio-demographic characteristics, including anthropometric measurements such as body mass index, waist circumference and body fat percentage, and the English and Malay version of the Global Physical Activity Questionnaire. A total of 261 primary healthcare workers participated in this study; 45.6% were classified as physically inactive, spending a median of five hours daily engaged in sedentary behaviours. In terms of metabolic equivalent min per week, male workers had significantly higher physical activity than females. Self-reported health status and longer sitting times were significantly associated with physical inactivity. Logistic regression showed that poor health status was 1.84 times less likely to be associated with physical activity (*p* = 0.036, Confidence Interval = 1.04–3.24). Due to the high prevalence of physical inactivity, action is needed to increase physical activity among healthcare workers.

## 1. Introduction

Physical activity is defined as any body movement by the skeletal muscles that causes energy expenditure [1]. Insufficient physical activity leads to an increased risk of lifestyle-related non-communicable diseases (NCDs), such as cardiovascular disease (CVD) and obesity. According to the World Health Organization (WHO), physical inactivity is the fourth leading cause of mortality globally, contributing to 6% of deaths annually [2]. Similarly, sedentary lifestyles play a significant role in the development of excess weight and obesity [3,4] and increase the risk of CVD, diabetes, colorectal cancer, hypertension, osteoporosis, lipid disorders, depression and anxiety [5].

To remain healthy and avoid the consequences of physical inactivity and sedentary behaviour, healthy people between 18 and 64 years of age should engage in at least 150 min of moderate-intensity physical activity or a minimum of 75 min of vigorous-intensity physical activity throughout the week [6]. Adopting a physically active lifestyle has been shown to improve cardiorespiratory fitness, muscular strength and bone health and to alleviate depression symptoms [6,7].

Despite these benefits, the level of physical activity is declining globally. The latest research shows that 28%, or 1 in 4 adults, are physically inactive [8]. Similarly, 33.5% of Malaysians were found to be physically inactive according to the National Health and Morbidity Survey (NHMS) 2015 [9]. Sedentary lifestyles due to urbanisation, work mode and modern transportation are factors in the rise of excess weight, obesity and physical inactivity [10,11]. Fewer places for recreational activity have also led to a substantial decrease in physical activity [12]. Healthcare workers are expected to be knowledgeable about healthy lifestyles and play a major role in public awareness of the consequences of physical inactivity. This is especially the case in primary care services, such as the government’s health clinics, which are among the most important places for patients to seek assistance from the medical care system and to educate the public on health or nutritional matters [13]. In Malaysia, the increasing number of patients visiting health clinics each year (approximately 40 million in 2016) reveals the importance of health clinics as the major healthcare setting for patients to seek treatments and advice [14].

There are various occupational classifications in the health industry, ranging from support staff, such as drivers, cleaners, clerical staff and accountants, to health professionals, such as doctors, nurses and laboratory technicians. All of them are classified into two categories: ‘health service providers’ and ‘health management and support workers’. Health service providers are directly involved in delivering health services and can be further divided into ‘professionals’ and ‘associates’. ‘Professionals’ includes doctors, pharmacists and other health professionals, whereas ‘associates’ consists of nurses, medical assistants and assistant pharmacists. Support staff, in contrast, are healthcare workers who are not directly involved in providing health services to the community. Instead, they act in a supporting role in order for the healthcare system to run smoothly. According to an International Standard Classification of Occupations report, ‘healthcare workers’ includes all the occupations listed within the health industry [15,16]. 

Unfortunately, there are still very few updated studies to determine the physical activity levels among healthcare workers. Nevertheless, considering the current literature, previous studies have shown a high incidence of physical inactivity among healthcare workers worldwide [17,18,19,20,21], with obese healthcare workers showing a higher likelihood of physical inactivity [22,23,24]. In addition, several studies also found an association between physical activity level and socio-demographic characteristics, such as age, marital status, occupational level and income. 

Iwuala et al.’s (2015) study found that older and married healthcare workers were less likely to engage in physical activity [23]. On the other hand, a recent Malaysian study found that medical staff and healthcare workers at the middle-income level were associated with physical inactivity. Non-medical staff, such as administrators and low-income groups, were found to be more engaged in physical activity, possibly due to the nature of their work, which required them to move around during working hours [25]. 

However, there were some studies that disagreed with the above conclusions, as their findings showed that most healthcare workers were physically active. For instance, two studies have been carried out in Malaysia, with both showing that most of the healthcare workers engaged in an active lifestyle and reporting no association between physical activity levels and BMI. A study was done in the UKM Medical Centre using the Global Physical Activity Questionnaire (GPAQ) to determine the levels of physical activity among healthcare workers. After analysis, 67.2% of the healthcare workers were categorised as physically active. The second study was done in health clinics in Hulu Langat, Selangor. It was reported that 64.5% of the healthcare workers were physically active when assessed using the International Physical Activity Questionnaire (IPAQ) [25,26].

Compared to the hospital setting, which is considered secondary care focusing on acute medical care, primary care services, such as health clinics, are more focused on disease management, maternal services and health promotion [13]. Thus, the nature of these jobs may be different from those of healthcare workers in hospital settings, as they are not required to move around frequently. This especially applies to doctors, who tend to sit in their rooms during working hours to treat patients, which may affect their physical activity levels. In addition, most studies have only focused on physical activity levels, neglecting the importance of looking into the sedentary behaviours among primary healthcare workers.

As such, this study sought to assess sedentary behaviours in addition to levels of physical activity in primary healthcare workers in Perak, and determine the association between physical activity levels and socio-demographic characteristics.

## 2. Materials and Methods

A cross-sectional study was conducted at 12 health clinics in three districts (Kinta, Kuala Kangsar and Perak Tengah) in the Malaysian state of Perak from August 2017 to April 2018. All primary healthcare workers aged 18 years old and above were invited to participate in this study. 

Each respondent was required to complete a self-administered questionnaire consisting of two sections. The first section collected socio-demographic characteristics including ethnicity, age, gender, monthly household income, marital status, educational level, occupational status and self-reported health status. Anthropometric measurements, including body mass index (BMI), waist circumference and body fat percentage of the respondents, were measured by the investigators at the health clinics. A SECA Portable Stadiometer and Omron HBF-375 were used to measure height, weight, BMI and body fat percentage, while waist circumference was measured with stretchable measuring tape.

The second section assessed respondents’ physical activity level using the Global Physical Activity Questionnaire (GPAQ) English and Malay version, validated by Soo et al. (2012) [27]. This questionnaire is aimed at determining the level of participation in physical activity in three domains: occupational physical activity, transport-related physical activity and physical activity during discretionary or leisure time [28]. The study also collected time spent on sedentary activities. Respondents were required to document time spent sitting or reclining during a typical day, which represented sedentary behaviour.

Metabolic equivalents (METs) were used to express the intensity of physical activity. This is the ratio of the work metabolic rate to the resting metabolic rate. One MET is expressed as 1 kcal/kg/hour and is equivalent to the energy required for sitting quietly. Moderately active people need four times higher calorie consumption, while vigorously active people require eight times higher. Thus, 4 METs were assigned to time spent on moderate activities and 8 METs to time spent on vigorous activities.

Physical activity levels were calculated by adding the total MET min of activity for each domain. Calculated activity levels were then classified as low, moderate or high [29]. The criteria for these three levels were as follows:High
⚬Seven or more days of any combination of the three domains AND total physical activity at a minimum of 3,000 MET min per week.⚬Three or more days of vigorous-intensity activities AND total physical activity at a minimum of 1,500 MET min per week.Moderate
⚬Three or more days of vigorous-intensity activities AND involving more than or equal to 60 min per week.⚬Five or more days of any combination of the three moderate-intensity activities AND involving more than or equal to 150 min per week.⚬Five or more days of any combination of the three domains AND total physical activity at a minimum of 600 MET min per week.Low
⚬The value does not meet the criteria for either high or moderate levels of physical activity.

Respondents were classified as ‘active’ if they accumulated a total of ≥ 600 MET min per week or were categorised as ‘moderate’ or ‘high’. Respondents failing to meet the total physical activity accumulation or being categorised as ‘low’ were classified as ‘inactive’ [29]. In addition, respondents were required to write down the time spent sitting or reclining on a typical day. The total time spent doing sedentary activities was split into two groups: ≤4 h and >4 h.

This study was approved by the National Medical Research Registry (NMRR), Medical Research and Ethics Committee (MREC) of the Ministry of Health Malaysia (Reference number: NMRR-16-2728-33440). Approval was also obtained from the Ethics Committee for Research Involving Human Subjects, Universiti Putra Malaysia (JKEUPM) (Reference number: UPM/TNCPI/RMC/1.4.18.2) and the State Health Department, Perak, Malaysia.

Collected data were analysed using IBM SPSS Statistics Standard Edition 20.0. Descriptive statistics were presented in terms of mean ± standard deviation (SD) and percentage for normally distributed variables. In the case of skewed distribution, medians with 25th and 75th percentiles were presented. Association between categorical variables was analysed via the Chi-square test. Logistic regression analysis was used to predict the probability of physical inactivity. All confidence intervals (CI) were set at 95% probability levels, and *p*-value < 0.05 was considered statistically significant.

## 3. Results

A total of 261 primary healthcare workers were recruited for this study. Consent was obtained from all primary healthcare workers before the study began. 

Overall, 45.6% (*n* = 119) of primary healthcare workers were classified as physically inactive. The workers spent a median of 1080 (240–3600) MET min per week on physical activities. Males had significantly higher physical activity levels than females, with a median of 2580 (870–7320) MET min per week and 960 (240–2880) MET min per week, respectively. When comparing the median time spent (in min) in different domains per week, it was found that all the primary healthcare workers only spent a median of 50 min per week engaged in occupational physical activity and a median of 80 min engaged in physical activity during leisure time. Males spent a significantly higher amount of time doing work-related and recreational physical activities, at 270 (0–1260) min and 151.5 (30–480) min per week more than females, respectively. However, neither males nor females were actively involved in transport-related physical activities; on average, neither group was found to be spending any time in this area. Meanwhile, 62.8% (*n* = 164) of primary healthcare workers spent over 4 h per day engaged in sedentary activities. The total sitting time, representing sedentary behaviours, was a median of 5 h daily. Male and female primary healthcare workers showed nearly the same sedentary behaviours, with median sitting times of 330 min and 300 min, respectively. Table 1 presents the physical activity levels and sitting times of the primary healthcare workers.

Details about socio-demographic characteristics and the anthropometric measurements of the primary healthcare workers are presented in Table 2. Most participants were Malay (80.8%), female (83.9%), below 40 years old (72.8%) and obtained tertiary education (83.1%). The majority of the primary healthcare workers were categorised as health service providers (80.1%), with more than half (66.3%) of them being ‘associates’, which includes staff nurses, medical assistants and pharmacy assistants. In addition, 75.1% self-reported that they were in good health. Looking at the anthropometric measurements, 49.9% of respondents were classified as overweight and obese, 51.0% were at risk of having abdominal obesity and 79.7% had high to very high body fat percentages. 

Table 3 shows the relationships between various characteristics and physical activity levels among study participants. A Chi-square test showed that self-reported health status (*p* = 0.034, x^2^ = 4.479) and sitting time (*p* = 0.034, x^2^ = 4.476) were significantly associated with physical activity level. Participants who self-reported good health status were found to be more physically active than respondents reporting poor health status. In addition, participants sitting for less than 4 h daily showed higher physical activity levels.

Table 4 shows the logistic regression analysis of various variables with physical activity levels among participants. Self-reported health status was the only variable that was significantly associated with physical activity level. It was found that participants with poor health status were 1.84 times less likely to engage in physical activity (*p* = 0.036, CI = 1.04–3.24). On the other hand, although occupational status showed no significant association with physical activity levels, comparison between groups showed that support workers are more likely to engage in physical activity compared to professionals groups (*p* = 0.025, CI = 0.15–0.88). 

## 4. Discussion

The benefits of physical activity are well documented, especially in preventing occurrence of NCDs, such as CVD, cancer, obesity and depression. Unlike the two other Malaysian studies, which reported over 60% of healthcare workers as being physically active [25,26], nearly half (45.6%) of the primary healthcare workers in the current study were found to be physically inactive. The current findings also show that the physical activity level of primary healthcare workers was lower than the general population as reported by the population-based NHMS 2015 (33.5%) [9].

Compared to the general population, in which only 33.5% of people were found to be physically inactive, a higher percentage of primary healthcare workers in this study did not actively engage in physical activity, yet they spent greater time in sedentary activities (median of five hours). This finding is consistent with other studies reporting that the majority of healthcare workers were physically inactive [17,19,21]. A possible explanation for low physical activity among primary healthcare workers includes the sedentary nature of their work [23,30].

As shown in Table 1, it was found that primary healthcare workers only spent a median of 50 min in occupational physical activity domains and did not engage in transport-related physical activity. As shown, female healthcare workers spent significantly less time in both occupational physical activity and physical activity in leisure time domains. Lack of places for recreational activities and family obligations also seem to be causes of physical inactivity, as most of the primary healthcare workers were married [23]. Although no significant association was found between occupations and physical activity levels in the current study, logistic regression did show that support workers were more likely to engage in physical activity than professionals and associates. This finding is consistent with a recent Malaysian study that showed that non-medical staff were more actively engaged in physical activity [25].

No significant correlation between most socio-demographic characteristics and physical activity levels was found, contradicting previous studies [23,25]; the number of primary healthcare workers with a low level of physical activity in each category was similar to those with a moderate or high level. Given these inconsistent findings, more studies are needed. It was found that self-reported health status and sitting time were the only two associations with physical activity level. Primary healthcare workers who claimed to be free of health problems were more active than their counterparts. In fact, some health problems—such as asthma, osteoarthritis, gout and CVD—may prevent them from engaging in physical activity, reducing opportunities for exercising [31]. Logistic regression analysis further showed that primary healthcare workers with a poor health status were twice as likely to be physically inactive.

Meanwhile, it was found that primary healthcare workers who tended to sit longer were significantly associated with a low physical activity level. In addition to the sedentary nature of their work, female dominance and the fact that most were married might explain this. This demographic may prefer family bonding time after work, such as playing with their children, thereby reducing their physical activity. 

As shown in the findings, female primary healthcare workers only spent a median of 80 min per week on leisure-time physical activity, whereas males spent a median of 151.5 min per week on leisure-time physical activity. While over half of study participants had a high BMI, waist circumference and body fat percentage, no association was found between physical activity levels and these obesity indices. According to Heinonen et al. (2013), obese individuals tend to prefer spending more time resting and sitting than being physically active [32]. The current study, however, did not find a similar trend. This contrasts with several other studies that found that obese healthcare workers tend to be more physically inactive than their normal-weight colleagues [23,33,34].

To our knowledge, this is the first study conducted among primary healthcare workers in Malaysia that assessed physical activity levels together with sedentary behaviours. Prior research mostly focused on physical activity alone in the hospital setting. The different setting in the current study can help to provide updated scientific knowledge on the physical activity levels of healthcare workers in the primary care setting. In addition, in contrast to most of the previous studies, this study reported on the time spent on domain-specific physical activities to provide a broader understanding of the engagement in physical activity. Therefore, the results can provide more reliable data and act as a reference for policymakers for future planning to improve the engagement rate with physical activity. It should be noted that all three domain-specific physical activities showed that the engagement rates were low among the primary healthcare workers, particularly in terms of transport-related physical activity. In addition, these findings also contribute to the global database of physical activity levels among healthcare workers. Currently, very few updated studies have been done among healthcare workers. Comparisons can be made between the current findings and prior studies conducted globally to determine potential ways and methods to improve the engagement rates 

Nevertheless, in the current study, most factors did not correlate with high physical inactivity levels among primary healthcare workers, contradicting the results of other studies. The inconsistent findings may be due to this study’s limitations relating to self-reporting of physical activity levels. Self-reported data carry the risk of under- or over-estimation of physical activity levels. Instead of determining physical activity with the GPAQ alone, future studies could use a pedometer as an objective tool to assess respondents’ physical activity levels. The results from both the pedometer and GPAQ can then be compared to provide more accurate findings.

## 5. Conclusions

This study found a high prevalence of physical inactivity among primary healthcare workers in Perak, Malaysia. In addition, high percentages of obesity and sedentary behaviours place them at increased risk of having NCDs. Poorer health status and longer sitting time were associated with low physical activity levels. 

Logistic regression showed that primary healthcare workers with a poorer health status were less likely to be engaged in physical activity. Intervention and a health promotion programme to promote engagement in physical activity is essential, as it can help to achieve healthy body composition and increase the productivity and performance of primary healthcare workers.

## Figures and Tables

**Table 1 ijerph-17-05947-t001:** Physical activity level and sedentary behaviours among the primary healthcare workers.

Physical Activity Level		Male *n* (%)	Female *n* (%)	Total *n* (%)
Total physical activity level (Metabolic equivalents (MET) min per week)	Inactive	14 (33.3)	105 (47.9)	119 (45.6)
	Active	28 (66.7)	114 (52.1)	142 (54.4)
	Median (25th–75th percentiles)	2580 (870–7320)	960 (240–2880) *	1080 (240–3600)
Setting specific physical activity (min per week)	Occupational physical activity, Median (25th–75th percentiles)	270 (0–1260)	40 (0–300) *	50 (0–435)
	Transport-related physical activity, Median (25th–75th percentiles)	0 (0–0)	0 (0–0)	0 (0–0)
	Physical activity during discretionary or leisure time; Median (25th–75th percentiles)	151.5 (30–480)	60 (0–180) *	80 (0–180)
Sitting time (hours per day)	≤4 h	16 (38.1)	81 (37.0)	97 (37.2)
	>4 h	26 (61.9)	138 (63.0)	164 (62.8)
	Median (25th–75th percentiles)	330 (180–555)	300 (180–540)	300 (180–540)

Independent t-test was performed, * *p*-value < 0.05 represents significance.

**Table 2 ijerph-17-05947-t002:** Socio-demographic characteristics and obesity indices among the primary healthcare workers.

Variables		Male *n* (%)	Female *n* (%)	Total
Ethnic	Malay	31 (73.8)	180 (82.2)	211 (80.9)
	Non-Malay	11 (26.2)	39 (17.8)	50 (19.1)
Age (years)	≤40	29 (69.0)	161 (73.5)	190 (72.8)
	>40	13 (31.0)	58 (26.5)	71 (27.2)
	Mean ± SD	37.1 ± 8.7	36.7 ± 7.9	36.8 ± 8.0
Educational level	Below Tertiary	8 (19.1)	36 (16.4)	44 (16.9)
	Above Tertiary	34 (80.9)	183 (83.6)	217 (83.1)
Marital status	Single	8 (19.0)	27 (12.3)	35 (13.4)
	Ever married	34 (81.0)	192 (87.7)	226 (86.6)
Monthly household income (RM)	≤3000	16 (38.1)	76 (34.7)	92 (35.2)
	3001–6000	18 (42.9)	105 (47.9)	123 (47.1)
	≥6001	8 (19.0)	38 (17.4)	46 (17.7)
Occupational status	Professionals	6 (14.3)	30 (13.7)	36 (13.8)
	Associates	25 (59.5)	148 (67.6)	173 (66.3)
	Support workers	11 (26.2)	41 (18.7)	52 (19.9)
Self-reported health status	Good	34 (81.0)	162 (74.0)	196 (75.1)
	Poor	8 (19.0)	57 (26.0)	65 (24.9)
Body Mass Index	Underweight (<18.5 kg/m^2^)	2 (4.8)	10 (4.6)	11 (4.2)
	Normal (18.5–24.99 kg/m^2^)	14 (33.3)	105 (47.9)	120 (45.9)
	Overweight (25–29.99 kg/m^2^)	17 (40.5)	57 (26.0)	74 (28.4)
	Obese (>30.00 kg/m^2^)	9 (21.4)	47 (21.5)	56 (21.5)
	Mean ± SD	27.1 ± 6.1	25.8 ± 5.5	26.0 ± 5.6
Waist circumference	Acceptable	20 (47.6)	108 (49.3)	128 (49.0)
	At risk	22 (52.4)	111 (50.7)	133 (51.0)
	Mean ± SD	93.2 ± 15.5	83.4 ± 12.9	85.0 ± 13.8
Body fat percentage	Normal	7 (17.1)	46 (21.0)	53 (20.4)
	High	34 (82.9)	173 (79.0)	207 (79.6)
	Median (25th–75th percentiles)	26.8 (23.2–29.7)	34.3 (30.9–37.7)	33.5 (29.0–37.2)

RM, Ringgit Malaysia. SD, Standard deviation.

**Table 3 ijerph-17-05947-t003:** Association between various characteristics with physical activity level.

Variables		Physical Activity Level		Value x^2^	*p*-Value
		Active *n* (%)	Inactive *n* (%)		
Ethnic	Malay	119 (56.4)	92 (43.6)	1.762	0.184
	Non-Malay	23 (46.0)	27 (54.0)		
Sex	Male	28 (66.7)	14 (33.3)	3.033	0.082
	Female	114 (52.1)	105 (47.9)		
Age (years)	≤40	106 (55.8)	84 (44.2)	0.539	0.463
	>40	36 (50.7)	35 (49.3)		
Educational level	Below Tertiary	26 (59.1)	18 (40.9)	0.468	0.494
	Above Tertiary	116 (53.5)	101 (46.5)		
Marital status	Single	14 (40.0)	21 (60.0)	3.382	0.066
	Ever married	128 (56.6)	98 (43.4)		
Monthly household income (RM)	≤3000	49 (53.3)	43 (46.7)	0.087	0.957
	3001–6000	68 (55.3)	55 (44.7)		
	≥6001	25 (54.3)	21 (45.7)		
Occupational status	Professionals	14 (38.9)	22 (61.1)	5.231	0.073
	Associates	95 (54.9)	78 (45.1)		
	Support workers	33 (63.5)	19 (36.5)		
Self-reported health status	Good	114 (58.2)	82 (41.8)	4.479	0.034 *
	Poor	28 (43.1)	37 (56.9)		
Body Mass Index	Underweight	5 (45.5)	6 (54.5)	0.876	0.831
	Normal	64 (53.3)	56 (46.7)		
	Overweight	40 (54.1)	34 (45.9)		
	Obese	33 (58.9)	23 (41.1)		
Waist circumference	Acceptable	70 (54.7)	58 (45.3)	0.008	0.929
	At risk	72 (54.1)	61 (45.9)		
Body fat percentage	Normal	32 (60.4)	21 (39.6)	0.956	0.328
	High	110 (52.9)	98 (47.1)		
Sitting time (per day)	≤4 h	61 (62.9)	36 (37.1)	4.476	0.034 *
	>4 h	81 (49.4)	83 (50.6)		

Pearson Chi-square test was performed. * *p*-value < 0.05, significant.

**Table 4 ijerph-17-05947-t004:** Logistic regression analysis of various variables associated with physical activity level among the primary healthcare workers.

Variable	OR	95% Cl	*p*-Value
**Ethnic** Malay Non-Malay	1 1.06	0.50–2.25	0.879
**Sex** Male Female	1 1.88	0.88–4.00	0.101
**Age** (**years**) ≤40 >40	1 1.43	0.76–2.71	0.269
**Educational level** Below Tertiary Above Tertiary	1 0.77	0.24–2.43	0.651
**Marital status** Single Ever married	1 0.46	0.20–1.07	0.070
**Monthly household income** (**RM**) ≤ 3000 3001–6000 ≥ 6001	1 0.92 0.96	0.54–1.59 0.47–1.95	0.957 0.768 0.904
**Occupational status** Professionals Associates Support workers	1 0.52 0.37	0.25–1.09 0.15–0.88	0.078 0.083 0.025 *
**Self-reported health status** Good Poor	1 1.84	1.04–3.24	0.036 *
**Body Mass Index** (**BMI**) Underweight Normal Overweight Obese	1 0.63 0.43 0.31	1.15–2.62 0.83–2.22 0.56–1.74	0.507 0.522 0.313 0.184
**Waist circumference** Acceptable At risk	1 1.25	0.54–2.91	0.603
**Body fat percentage** Normal High	1 1.72	0.77–3.80	0.181
**Sitting time** (**per day**) <4 h >4 h	1 1.53	0.87–2.69	0.141

* *p*-value < 0.05, significant.

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
