# Peer review of "Level of Physical Activity and Its Associated Factors among Primary Healthcare Workers in Perak, Malaysia"

_ijerph, 2020, doi:10.3390/ijerph17165947_

Round 1
Reviewer 1 Report
This research analyzes the self-reports of physical activity of health care workers in 12 health clinics in Malaysia.
The GPAQ is a self-report on the presence of MVPA, active transport and sitting in the sampled health worker population. It is true that it directly addresses the standard for cardio-pulmonary fitness but fails to include occupational walking and other physical efforts while working. In spite of the authors' suggestion that in the clinic environment most health workers work seated, it seems to me they have to move around a fair bit to do their jobs. I have now read some of the cited works, including especially Chin et al., (2016), but still did not find anything on activity related to the occupation.
As reported in the literature, senior managers and frontline staff exhibit more sedentary behaviours than staff nurses, for example. I wonder if this distinction can be made in the sample. Also, overall sedentariness may be related to self-perception of the job as physically and mentally demanding.
Although the authors suggest that the health outcome is related to sedentariness in the work environment and then outside, we also do not know if this is a consequence of the work environment itself, socio-affective state of the workers or other social characteristics of the workers. Do women often have domestic duties that leave them little time to engage in any leisure-based physical activity? But the domestic duties themselves are probably involving moving and doing, if not MVPA. Finally, is obesity related to any of the above or is primarily a consequence of diet?
Reviewer 2 Report
Thank you for this interesting manuscript on the Level of Physical Activity and its Associated Factors Among Primary Healthcare Workers in Perak, Malaysia.
I enjoyed reading the manuscript, but a few things were not entirely clear in the present version. Throughout the manuscript you change between indicating what makes primary healthcare workers less likely to be physically active, more likely to be physically active, more inactive, and less inactive. This makes it quite hard to get your head around how the different variables relate. It would be clearer to pick one way to phrase the outcome variable, and relate everything to that. Either the likelihood of being active OR inactive, and express everything as being more likely to achieve the outcome, OR less likely. Below are specific points, by line number. I would like to see more detail in the tables, more information about the data you collected, including the difference between work time and leisure time PA and sedentary time. I think you can back up some of your arguments with data you do have, but have not reported on, which would greatly strengthen your arguments. I would like you to pay particular attention to the final comment, as you are reporting reversed directionality in your conclusion, which is very problematic. However, most of my other points are very minor, and if all points below are addressed, I expect this to be a very good paper.
Line 43-45: You say 33.5% of Malaysians were found to be inactive, with 4 out of 10 categorised as physically inactive. This states the same thing twice in the same sentence (physical inactivity), but I can't equate 4 out of 10 with 33.5%.
Line 56-58: 'non-medical staff and low-income groups' You link these groups to having more physically active work than primary healthcare workers. I would expect nurses (especially in hospitals) to be very physically active at work. Later in the manuscript (discussion, line 168-170) you describe doctors do a lot of their work sedentary. It would be beneficial to have this information mentioned with lines 56-58.
Line 64-71: did participants take their own height and weight measures? Was this done by research staff? Were they visited at home? At work?
Line 83 onwards: the description of PA minutes is very confusing. Some are expressed as minutes, others as MET minutes. I can't work out what MET minutes are. The categories are high, moderate, or low, but the remainder of the manuscript discusses active or inactive. Is low the same as inactive? Is moderate or high the same as active?
Line 108: you say the descriptives are presented as mean and standard deviation, but there is no table that presents these descriptives. It would be highly informative to have this data for all variables. Currently only the number of participants in each category is presented in the manuscript.
Line 116: what does the category health service provider actually mean? Which professions are included in this?
Line 120: "and sitting time </= 4h (43%)". This sentence would make more sense if 'and' was replaced with 'or' as respondents belonging to any one of the categories was found to be more physically active, rather than them needing to belong to all categories at the same time.
Table 1: there is no mention anywhere of what this table displays. I guess it's number of respondents and percentages? The physical activity labels at the top of the columns are confusing - in the text you talk about active vs. inactive, so that should also be the labels for the columns, rather than moderate/high vs. low. I am missing information on the mean/SD or percentiles here. Also, the categories for marital status are 'single' and 'ever married' - I am not sure how to interpret 'ever married', and later in the text you refer to healthcare workers who are married as being less active. Are they married? Or are they married or were so at some point in their life?
Line 127: There is no explanation in the manuscript of the category 'physically inactive'. There is an explanation for low physical activity - is this the same?
Line 128-130: I have not been able to understand from previous paragraphs what these MET minutes mean so I have no idea if 1080 is a lot or very little. Earlier you mentioned 600 being the cut for moderate activity, so this seems high. Also, you report the median, but no percentiles, so I have no idea how this is distributed across the sample. As you report the median rather than the mean, I assume this is a skewed distribution?
Line 131-132: You are mixing time expression in hours (4 hours) and minutes (300 minutes).
Table 2: * Significant. What is significant? What test have you carried out? What is the test statistic that is significant? At the top of the columns you indicate 'n (%)'. However, this doesn't apply to the median scores. These also need an explanation of what they are.
Line 140-141: You have not reported the chi-square statistic, only the significance level. You then claim a strong correlation - please report the chi-square statistic.
Table 3: Please report the chi-square statistic. The star used to indicate significance is not explained in the note at the bottom. Please indicate what the data in the columns is (I guess it's n and % again?).
Line 147: logistic regression does not assess correlation, it investigates predictive association.
Table 4: star for significance missing from note at the bottom.
Line 157-158: "nearly half of ... be physically inactive." There is no table in the manuscript that displays these numbers clearly. In all tables, the sample is further split into categories that make it really difficult to get an overview of the data.
Line 162: "a higher percentage of primary healthcare workers do not actively engage" should say 'of primary healthcare workers in this study'.
Line 166-168: you refer to the low level of PA in work time and due to being married and spending time with family. It would be really interesting here to use the fact that your questionnaire separately assessed PA at work, for transport, and during leisure time. If you want to relate being married to being less active due to spending time with family, this would impact ONLY on leisure time PA, so you should be able to show this with data. If you want to make an argument for healthcare workings being sedentary at work, you again have the data to do this. Making these suggestions without backing it up with data, when you do have the data available does not give you any credibility.
Line 173: lack of correlation with socio-demographic characteristics - this may be related to the cutpoints chosen and the dichotomisation of the variables. Some of these would perhaps be more informative as a continuous or categorical variable. What is the theory behind the dichotomisation and the cutpoints chosen?
Line 183-184: It would be good to have more information here about whether this sedentary time is in work time or also across leisure time. Does this mean they do not compensate for sedentary work by having active leisure time? Do sedentary time at work and during leisure correlate? Or sedentary time at work and PA during leisure time?
Line 185-186: As above, if you want to explain marital status as inflluencing PA it is important to look at leisure time PA specifically.
Line 191-192: I can not find any data in the results section relating sitting time to BMI or body fat percentage. If you are reporting this in the discussion, it needs to be in the results section, backed up with numbers.
Line 197: "reference to policy makers for future planning". This is a very good point, but for future planning of what? Leisure facilities? Then you need to make an argument - backed up with data - that leisure time PA is particularly low, or not high enough to compensate for sedentary working life. For active transport capacity? Then you need to back up that transport PA is low or non-existent and this must be addressed.
Line 200: "most factors did not correlate" I think this may be to do with how you manipulated the variables (dichotomisation of all variables) rather than there being a lack of correlation. Especially obesity and body fat tend to only have a noticeable effect once people are very obese, not just overweight (BMI >30 rather than 25, and those who are underweight (BMI <20) are also often less active, which is now mixed in with your 'healthy BMI' category.
Line 211-212: "Logistic regression showed that..." This is absolutely NOT what your logisitic regression showed. Your outcome variable was physical activity, and this sentence reverses the directionality of the test you performed. What you determined is that those with lower health status were less likely to be physically active, but you cannot conclude the opposite directionality from the logistic regression you performed. In that case you would need to use health status as the outcome variable.

Reviewer 3 Report
Since there have been many cross-sectional studies describe physical activity levels among adults in Malaysia, why the authors conducted this study?
The conclusions have already been known from the literature.
What is the difference between this study and others?
Thus, the innovativeness, need, and importance of this study are not clear.
Reviewer 4 Report
This paper associates the level of physical activity and its associated factors in primary healthcare workers in Perak, Malaysia. The manuscript is interesting; however, authors should address come concern prior publication. Please, see my specific comments below.
- The manuscript seems a very regional manuscript and I’m not sure how the results can improve the current knowledge about the physical activity in healthcare workers worldwide. In discussion section there is no clear global benefit from the results.
- Page 7, lines 195-196: “To our knowledge, this is the first study conducted among primary healthcare workers in Perak, Malaysia that assessed physical activity level together with sedentary behaviours.” The manuscript presentation just highlighted importance of the manuscript in Malaysia as a regional study.
- In the end of the introduction section there is no study aims or goals described clearly.
- Authors performed univariate analysis to identify different association, however, no multivariate analysis was performed. This reviewer believes that a multivariate analysis could represent better the association among all characteristics.
- The conclusion section is just repeating the results and did not shown any important conclusion (related to the results).
Round 2
Reviewer 1 Report
I have no additional comments. I believe the GPAQ can easily lead to erroneous conclusions but at least the tendencies are fairly represented in this population.
Reviewer 3 Report
The authors provided a revised version of the manuscript including all recommendations asked.
No further comments.
Reviewer 4 Report
The authors provided a revised version of the manuscript including all recommendations asked. The manuscript text changed substantially including all required information. This reviewer has no further comments.
This manuscript is a resubmission of an earlier submission. The following is a list of the peer review reports and author responses from that submission.